# Combining 3D U-Net and bottom-up geometric constraints for automatic cortical sulci recognition

**Léonie Borne**          LEONIE.BORNE@CEA.FR

**Jean-François Mangin**          JEAN-FRANCOIS.MANGIN@CEA.FR

**Denis Rivière**          DENIS.RIVIERE@CEA.FR

*Neurospin, CEA Saclay, 91191 Gif-sur-Yvette, France*

## Abstract

While the limits of deep learning are still to be clarified, some problems may benefit from a mixed approach combining CNNs with traditional strategies. For instance, bottom-up representations embedding domain-specific knowledge could help to regularise a voxelwise segmentation. In this paper, we propose such an approach dedicated to the automatic recognition of the human cortical sulci, designed as the labelling of the voxels of a skeleton of the fluid surrounding the brain. Deep learning is used to provide a top-down perspective to a classical bottom-up pattern recognition system. Our original approach is compared with the approach proposed in the BrainVISA package (www.brainvisa.info), the most used sulcus recognition toolbox. As far as we know, this is the first time that CNNs is used for sulcus recognition. We show that our approach outperforms the BrainVISA method.

**Keywords:** U-Net, geometric constraints, cortical sulci.

## 1. Introduction

The cortical surface is made up of many convolutions, called gyri, delimited by folds, called sulci. The main sulci provide a kind of road map delimiting functionally different regions. The shape of the sulci is used as biomarkers of developmental and neurodegenerative diseases. Despite the many tools available to visualize sulci in 3D, their labelling is long and fastidious. However, because of the large variability of the folding pattern in the general population, inferring developmental biomarkers requires the mining of thousands of brains, so the automation of sulci recognition is essential. As in (Perrot et al., 2011; Lohmann and Von Cramon, 2000; Destrieux et al., 2010), in order to facilitate the task, sulci recognition can be divided into two subtasks: the segmentation of cortical folds and their labelling.

In this study, we propose to use the 3D U-Net architecture described in (Beers et al., 2017) for labelling, based on the robust and already widely used fold segmentation provided by the BrainVISA toolbox. However, in general, fully connected networks as U-Net treat voxels independently of each other. Thus, there is no guarantee that the geometric definition of a sulcus, as a set of topologically simple surfaces, is respected. This is particularly disabling for morphometric studies whose measurements are based on this definition. In order to remedy this, bottom-up geometric constraints are proposed. The new model is compared to the BrainVISA model (Perrot et al., 2011), based on the same fold segmentation.

## 2. Method

Thanks to the BrainVISA pipeline, already widely used for studying cortical anatomy, the folds are represented by a set of voxels corresponding to a skeleton of the cerebrospinal fluid filling the fold. This representation of the folds can be understood as a negative mold of the brain. After this data preprocessing, the labelling is done in two steps: first, the labelling of skeleton voxels using a 3D U-Net, second, the spatial regularization of the results.

### 2.1. Skeleton voxels labelling thanks to 3D U-Net

The fold skeleton is registered in Talairach space and used as input: it corresponds to a 3D binary volume with a common resolution of 2*2*2mm, where the voxels belonging to the skeleton are one and the others are zero. In order to augment the training data, a rotation in a random direction with a random angle (following a Gaussian distribution $\mathcal{N}(0, \frac{\pi}{16}^2)$) is applied to the images at each epoch.

Stochastic gradient descent is used for training, with learning rate and momentum determined by 3-folds inner cross validation. The learning rate is halved when the loss function has not improved for two consecutive epochs. After four consecutive epochs without improvement, training is stopped. The loss function used is the cross entropy loss. As the voxels that do not belong to the sulci skeleton, i.e. a large majority of the voxels in the image, do not need to be classified, these voxels are not used for gradient backpropagation. The model obtained permits to label the skeleton at the voxel scale (U-Net).

### 2.2. Bottom-up geometric constraints

BrainVISA pipeline is also providing a clustering of the skeleton voxels into elementary folds, the building blocks of cortical morphology. The straightforward approach to regularize the results is to do a weighted majority vote: for each elementary fold, the scores are averaged by label and the highest score label is kept (U-Net + Reg.). However, the fragmentation of the skeleton into elementary folds is not robust: even from the same MRI, very different fragmentation can be obtained because of stochastic optimizations embedded in the pipeline.

In this study, we propose to redivide the elementary folds thanks to the Ward's hierarchical agglomerative clustering method (Ward Jr, 1963) (U-Net + Reg. + Cut.). Clustering is performed for each elementary fold, based on the scores obtained for each label. In order to ensure spatial consistency, a spatial connectivity constraint is imposed during cluster agglomeration. Then, the Calinski-Harabaz index (Caliński and Harabasz, 1974) is used to quantify the quality of the proposed clustering. If this score is higher than a threshold determined by inner cross validation, the partitioning is performed. When an elementary fold is split in two, each of the two clusters obtained are also challenged with the same manipulation, until all the elementary folds have a Calinski-Harabaz index below the threshold.

## 3. Experiment

### 3.1. Database

The training base is composed of 62 healthy brains selected from different heterogeneous databases and labelled with a model containing 63 sulci for the right hemisphere and 64 for

the left. Each elementary fold, extracted thanks to the BrainVISA pipeline, were manually labelled and have been manually cut out when needed. Compared to (Perrot et al., 2011), the database is the same but four additional sulci were used.

### 3.2. Error rate

Error rates are assessed by 10-folds cross validation. One model is trained per hemisphere.

As in (Perrot et al., 2011), the $E_{SI}$ error rate is used to evaluate the model. However, in order to take into account the variability of the fragmentation into elementary folds and therefore the robustness of the labelling methods to this variability (thanks to bottom-up geometric constraints), each image was re-segmented ten times. Thus, if the image belongs to the training set, only the segmentation used for manual labelling is considered. However, if the image belongs to the test set, ten other segmentations (whose true labels have been transferred from manual segmentation) are labelled and used to quantify the $E_{SI}$. For each subject, from the ten segmentations, the average of the errors, $E_{SI}^{mean}$, and the maximum error, $E_{SI}^{max}$, are considered. Note that in (Perrot et al., 2011), only manual segmentation is used to evaluate the $E_{SI}$ in leave-one-out, hence the results presented here are worse.

## 4. Results

The comparison of U-Net + Reg. + Cut. and BrainVISA models with a matched T-test indicates that the average $E_{SI}^{mean}$ and $E_{SI}^{max}$ (Table 1) are significantly improved ($p_{value}$=4.50e-17 and 8.39e-20). In addition, regularization by elementary folds significantly improves the results, and their automatic re-division also allows a significant improvement.

Table 1: $E_{SI}^{mean/max}$ ($2\sigma$) in % and $p_{value}$ of the T-test comparing to the previous line model.

| **Error**/$p_{value}$ | $E_{SI}^{mean}$ | $p_{value}$ | $E_{SI}^{max}$ | $p_{value}$ |
|---|---|---|---|---|
| (Perrot et al., 2011) | 18.45 (6.02) | | 20.79 (6.64) | |
| U-Net | 16.85 (4.67) | **5.88e-6** | 17.68 (4.83) | **3.10e-14** |
| U-Net + Reg. | 15.54 (5.47) | **2.94e-14** | 17.47 (5.92) | 2.11e-1 |
| U-Net + Reg. + Cut. | **15.06 (5.10)** | **9.80e-6** | **16.72 (5.40)** | **5.44e-8** |

## 5. Conclusion

In summary, despite the high variability of the folds, making it particularly challenging to automate their recognition, this study once again shows the power of CNNs compared to the methods developed so far. Compared to (Perrot et al., 2011), the proposed model is significantly better. But above all, it is robust to errors in elementary folds fragmentation. Note that despite the definition of elementary folds specific to this problem, defining a coherent geometric entity for a given segmentation problem is already widely used, for example by using super-pixels (Giraud et al., 2017; Soltaninejad et al., 2017) that group the most similar connected pixels together so that they have the same label.

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
