# OpenReview forum: "Combining 3D U-Net and bottom-up geometric constraints for automatic cortical sulci recognition"
_MIDL.io/2019/Conference/Abstract — MIDL Abstract 2019_

### Official Review · AnonReviewer1 · 2019-04-26
**this work has discussion potential and shows reasonable results**

**Rating:** 3
**Confidence:** 2

**Review:**

The abstract discussed the combination of a U-Net with geometric constraints for the segmentation of e human cortical sulci.

This is an interesting work with only some minor weaknesses:
- even-though the U-Net is well known by now, Ronneberger should be cited.
- would this approach still work when pathologies are present or if the cortex deviates a lot from adult brains, e.g. developing fetal brains?
- I assume 'T-test comparing to the previous line model.' means the p-value always comparing to the previous row in the table?

this work has discussion potential and shows reasonable results.

---

### Official Review · AnonReviewer2 · 2019-04-29
**Hierarchical agglomerative clustering as a post-processing step for CNN segmentations**

**Rating:** 3
**Confidence:** 1

**Review:**

* A 3D U-net is trained for segmentation and spatial regularization is imposed as a post-processing step via hierarchical agglomerative clustering.
* Some more details would be useful. For instance, what type of spatial connectivity constraint is imposed during cluster agglomeration?
* The experiments seem fairly extensive. I especially like the relatively high number of cross-validation folds.
* Some qualitative results would be helpful for those not familiar with the images being dealt with in this work.

---

### Decision · Program_Chairs · 2019-05-06
**Acceptance Decision**

Accept